# Alleviating Effect of *Lactiplantibacillus plantarum* NXU0011 Fermented Wolfberry on Ulcerative Colitis in Mice

Mingxia Nie [1,†], Quan Ji [1,†], Gang Guo [2,‡], Haiyan Zhang [1], Yanhong Wang [1], Ru Zhai [1] and Lin Pan [1,*]

[1] Ningxia Key Laboratory for Food Microbial-Applications Technology and Safety Control, Food Science and Engineering College, Ningxia University, Yinchuan 750021, China; xia2374109642@163.com (M.N.); jq18295154870@163.com (Q.J.); yanhong1789@126.com (Y.W.)
[2] Ning Xia Eppen Biotech Co., Ltd., Yinchuan 750021, China
[*] Correspondence: panlin@nxu.edu.cn; Tel.: +86-13674887257
[†] These authors contributed equally to this work.
[‡] This author made secondary contributions to this research work.

**Abstract:** As research into the relationship between the gut microbiome and health continues to evolve, probiotics are garnering increasing interest among consumers. Fermentation is recognized as an efficacious biotechnology for augmenting the nutritional and functional attributes of foods. In this study, the ameliorative effects of *Lycium barbarum* L. lyophilized powder fermented with *Lactiplantibacillus plantarum* NXU0011 (LP+*Ly*) on dextran sodium sulfate (DSS)-induced ulcerative colitis (UC) in mice were investigated employing immunohistochemistry, qRT-PCR, macrogenomics, and metabolomics. The results revealed that LP+*Ly* intervention significantly ameliorated histopathological inflammation in the ulcerated colon, diminished the expression of inflammatory markers such as IL-6, P-STAT3, and miR-214, and enhanced the diversity of intestinal flora in the mouse model group. Moreover, there was an increase in the abundance of beneficial bacteria, including *Lactobacillus*, *Prevotella*, and *Akkermansia*. Metabolomic analysis indicated that 15 metabolites, including citrulline, D-xylose, and α-ketoisovaleric acid, exhibited significant variations following the LP+*Ly* intervention. The metabolic pathways that displayed substantial differences included tryptophan biosynthesis, arginine biosynthesis, and amino sugar and nucleotide sugar metabolism. LP+*Ly* effectively improved the inflammatory state within the intestines by modulating arginine biosynthesis, thus alleviating the impact of UC.

**Keywords:** Probiotics fermentation; *Lycium barbarum* L.; ulcerative colitis; immunohistochemistry; macrogenomics; metabolomics

## 1. Introduction

Probiotics are living microorganisms. When consumed in appropriate amounts, they can provide health benefits to the host, including protecting the intestines, anticancer effects, immune regulation, lowering blood lipids, controlling blood sugar, and promoting the absorption of nutrients. In the treatment of Ulcerative colitis (UC), potential mechanisms include inhibiting the growth of pathogenic bacteria, improving epithelial mucosal barrier function, regulating immunity, and reducing the secretion of pro-inflammatory factors [1–3]. *Lycium barbarum* L. (L), belonging to the Solanaceae Lycium genus, is renowned for its antioxidant, anti-inflammatory, and anti-cancer properties [4]. UC is a type of inflammatory bowel disease (IBD) that begins in the rectum and typically extends proximally to involve the entire colon. Its main pathological manifestations include intestinal barrier damage, loss of crypts, crypt abscesses, deformation of mucosal glands, infiltration of inflammatory cells, and loss of goblet cells. Currently, aminosalicylate drugs, including SASP and mesalazine, are commonly used for the treatment of UC. However, these drugs have potential adverse reactions such as drowsiness, gastrointestinal complications, and renal dysfunction. L-arginine plays a role in immune regulation and nitric oxide synthesis

in UC [5–7]. In this study, probiotics were combined with L to create a lyophilized bacterial powder through fermentation, which exhibits potential probiotic efficacy. Dextran sodium sulfate (DSS)-induced UC in mice was used as a model to investigate the effects of fermented L lyophilized powder (LP+*Ly*) on the intestinal flora and metabolite profile. This study integrated immunohistochemistry, quantitative reverse-transcription PCR (qRT-PCR), macrogenomics, and metabolomics to explore the potential therapeutic value of LP+*Ly* and provide new insights for the prevention and alleviation of UC.

## 2. Materials and Methods

### 2.1. Preparation of LP+Ly

The dried fruits of L were soaked and pulped in pure water at a mass ratio of 1:5 with 0.05% sodium D-isoascorbate for color protection. Allow it to soak at room temperature for 8–9 h. Then, add 0.15% pectinase and blend. Filter out the seeds using three layers of sterile gauze and a funnel. Next, add 5% white sugar and 10% citric acid. with a pH of 4.5, followed by pasteurization. The sterilized and cooled goji juice was inoculated with 5% (*v/v*) of *Lactiplantibacillus plantarum* strain NXU0011 and fermented at 37 °C for 12 h. It was then subjected to vacuum freeze-drying for 30 h and ground to obtain the final product, denoted as LP+*Ly* [8].

### 2.2. Animal Experimental Design

Thirty male SPF-grade C57BL/6J mice aged 6–8 weeks and free of specific pathogens were randomly divided into five groups: control (Con) group, model (DSS) group, positive drug (MS) group, LP+*Ly* high-dose (High) group, and LP+*Ly* low-dose (Low) group. The mice were acclimated and fed for 3 days. The Con group was given sterile water, while the other groups received sterile water containing 5% DSS (*w/v*) for 7 days to induce UC. Subsequently, the Con and DSS groups were administered 0.2 mL/day of phosphate-buffered saline (PBS) via gavage, the MS group received mesalazine dissolved in PBS at 40 mg/kg, and the High and Low groups were given $2 \times 10^9$ CFU/mL and $2 \times 10^6$ CFU/mL, respectively, of viable bacteria dissolved in PBS. On day 11, all mice were euthanized, and blood, colonic tissue, and feces were collected. The animal experiments were approved by the Experimental Animal Ethics Committee of Sichuan Lilaisinuo Biological Technology Co., Ltd. (Approval No.: LLSN-2022014).

### 2.3. Disease Activity Index (DAI)

During the modeling phase, the body weight, fecal characteristics, and fecal bleeding symptoms of each group of mice were observed and recorded daily. The DAI score was calculated by dividing the total score of the above three parameters by 3 [9]. The scoring criteria are provided in Table A1.

### 2.4. Histological Assessment

The colonic length was measured in all animals, followed by fixation in a 10% formalin solution for 48 h. The samples were dehydrated, embedded in paraffin, sectioned, and stained with hematoxylin and eosin (H&E). Microscopic observations were conducted to assess the pathological changes, and images were captured. The histological evaluation was based on the severity of inflammation, crypt damage, and the extent of pathological alterations [10,11].

### 2.5. RNA Isolation and qRT-PCR

RNA from colon tissue was extracted using TRIzol, and cDNA was synthesized through reverse transcription. qRT-PCR was performed using the Ultra SYBR Mixture reagent kit, with GAPDH as the internal reference gene. The primer sequences are provided in Table 1.

**Table 1.** The sequences of qPCR primers.

| Gene | Forward (5′-3′) | Reverse (5′-3′) |
|---|---|---|
| miR-214 | CGCTTTACAGCAGGCACAGA | TAAGGTTCATCACGACAGGTICAC |
| GAPDH | AAGGTCGGAGTCACCGGATT | CTGGAAGATGGTGATGGGATT |

The reaction system was prepared with a total volume of 20 μL, as described in Table A2. Gene expression levels were determined using the relative quantification method, specifically the $2^{-\Delta\Delta Ct}$ method.

### 2.6. Immunohistochemistry Staining

Paraffin sections of the colon tissue from each experimental group were dried in a constant-temperature oven at 60 °C for 1 h. The sections were then washed three times with 0.01 mol·L$^{-1}$ PBS solution for 5 min each, followed by washing with absolute ethanol for 10 min and two rinses with distilled water. Antigen retrieval was performed by heating the sections at 100 °C for 8 min, and the sections were then treated with 3% hydrogen peroxide for 30 min. The sections were incubated overnight at 4 °C with p-STAT3 and IL-6 antibodies, followed by incubation with a secondary antibody. DAB staining was performed, followed by counterstaining with hematoxylin. The sections were observed under a microscope, and micro-images were captured at 400× magnification.

### 2.7. DNA Sequencing and Gut Microbiota Analysis

Total microbial genomic DNA samples were extracted using the OMEGA Mag-Bind Soil DNA Kit (M5635-02; Omega Bio-Tek, Norcross, GA, USA) according to the manufacturer's instructions and stored at −20 °C for further analysis. The quantity and quality of the extracted DNA were assessed using a Qubit™ 4 Fluorometer (Invitrogen, Waltham, MA, USA) and agarose gel electrophoresis, respectively. The microbial DNA was then processed to construct metagenome shotgun sequencing libraries with 400-bp insert sizes using the Illumina TruSeq Nano DNA LT Library Preparation Kit. Each library was sequenced on the Illumina NovaSeq platform (Illumina, San Diego, California, USA), employing a PE150 strategy. The resulting sequencing data were analyzed to study the composition and functional changes of the gut microbiota among different groups.

### 2.8. Plasma Metabolites Analysis via Untargeted Metabolomics

Fasting blood samples were collected into 5-mL Vacutainer tubes containing the chelating agent EDTA and centrifuged at 1500× *g* and 4 °C for 15 min. Aliquots (150 μL) of plasma were stored at −80 °C until analysis. The plasma samples were thawed at 4 °C, and 100-μL aliquots were mixed with 400 μL of cold methanol/acetonitrile (1:1, *v/v*) for protein precipitation. The mixture was centrifuged at 14,000× *g* and 4 °C for 20 min. The supernatant was then dried in a vacuum centrifuge and re-dissolved in 100-μL acetonitrile/water (1:1, *v/v*) for UHPLC-Q-Exactive Orbitrap mass spectrometry (MS) analysis. The samples were finally centrifuged at 14,000× *g* and 4° C for 15 min before injection.

### 2.9. Statistical and Bioinformatics Analysis

Normalized data were analyzed using GraphPad Prism software (ver. 8.0.2; GraphPad Software Inc., San Diego, CA, USA). Model stability was evaluated using 7-fold cross-validation. The significance of differences between two independent samples was determined using a *t*-test, with significance criteria of variable importance projection (VIP) > 1 and $p < 0.05$.

Quality-filtered raw sequencing reads from the metagenomic data were classified for each sample using Kraken2. CDS sequences from all samples were clustered with mmseqs2 at a protein sequence similarity threshold of 0.90. Linear discriminant analysis effect size (LEfSe) was employed to detect taxonomic and functional differences between

groups and identify taxa and functions significantly enriched in different groups. Bray-Curtis distance was used to measure the dissimilarity in microbial community composition and function among samples. Principal coordinate analysis (PCoA) was employed for visualizing differences and similarities between samples based on microbial profiles.

The raw MS data were converted to mzXML format using ProteoWizard MSConvert. The converted data were processed with XCMS software for data extraction. Metabolite structure identification was conducted using a standard compound database. Both univariate and multivariate statistical analyses were used for intergroup differences and differential metabolite analyses. Significantly different metabolites were subjected to cluster analysis, Spearman correlation analysis, and bioinformatics analysis involving Kyoto Encyclopedia of Genes and Genomes (KEGG) pathways.

## 3. Results

### 3.1. Protective Effect of LP+Ly on DSS-induced Colitis in Mice

As depicted in Figure 1, throughout the UC model induction phase, the severity of the disease escalated, resulting in an increase in the DAI. Treatment with DSS significantly reduced body weight (Figure 1A) and colon tissue length (Figure 1C). Contrasting with the DSS group, supplementation with LP+*Ly* and the medication significantly alleviated changes in body weight and colon tissue length, with the mitigating effect being greatest in the Low group.

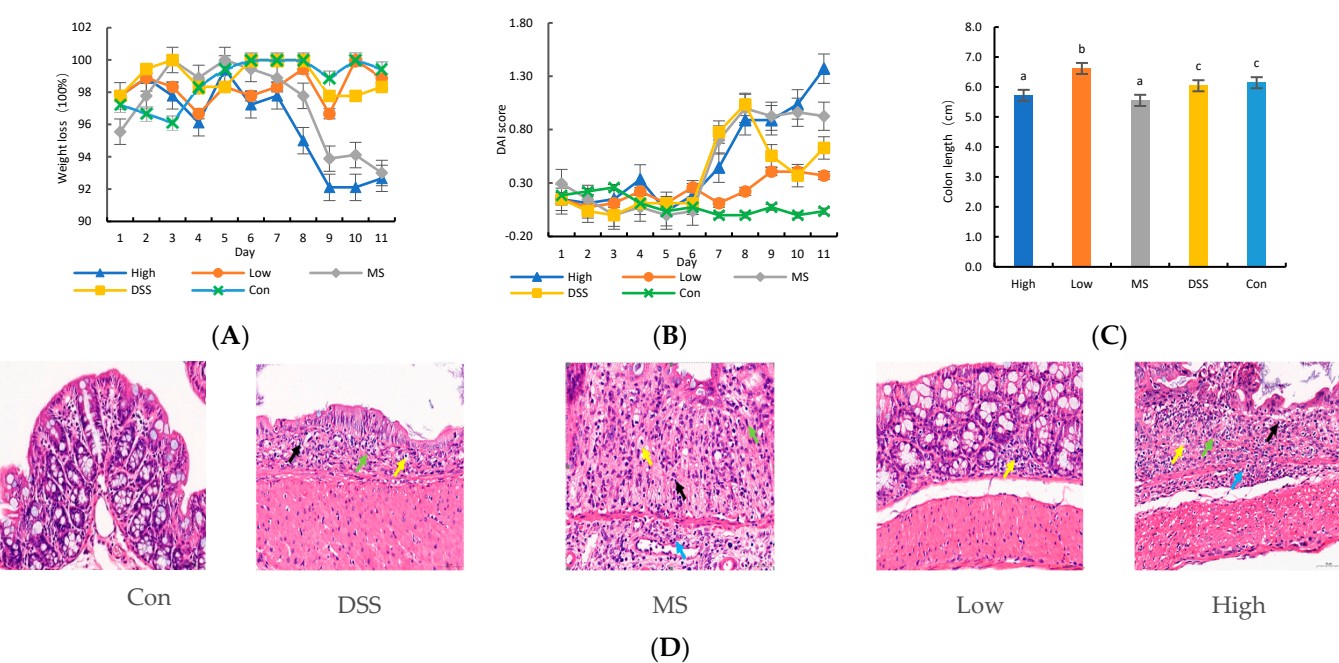

**Figure 1.** Effect of LP+*Ly* on the symptoms of colitis in UC mice. (**A**): The body weight changes. (**B**): DAI. (**C**) The colon tissue length (different letters indicate significant differences, $p < 0.05$). (**D**) typical H&E stained tissue sections in representative mice from each group (×400, scale bars: 5 μm, at the arrow: ulcerative focus (↑), necrotic enteric gland (↑), neutrophil (the most common type of white blood cell) (↑), lymphocyte (↑), fibroblast (↑)).

H&E staining (Figure 1D) revealed that the DSS group, in comparison to the Con group, exhibited colonic epithelial necrosis, mucosal integrity disruption, loss of goblet cells, and infiltration of inflammatory cells within the mucosal layer. The LP+*Ly* group and the medication-supplemented group both demonstrated significant improvement in colonic ulcer symptoms compared to the DSS group, with the Low group showing the most significant improvement. Further validation will be conducted to evaluate the therapeutic effect of low-dose LP+*Ly* on UC in mice.

### 3.2. Effect of LP+Ly on Cytokine Concentrations in the Colon

The expression levels of the miR-214 gene, IL-6, and p-STAT3 protein were assessed using qRT-PCR, and the results are presented in Table 2.

**Table 2.** Expression changes of cytokines in mouse colon tissue (±SD).

| Group | Number of Samples (Cases) | miR-214 | IL-6 | P-STAT3 |
|-------|---------------------------|---------|------|---------|
| Con | 6 | 1.147 ± 0.768 | 0.8007 ± 0.3948 | 0.9345 ± 0.9819 |
| DSS | 6 | 3.623 ± 1.128 ## | 6.2169 ± 1.8290 ## | 17.3432 ± 5.2125 ## |
| MS | 6 | 1.197 ± 0.686 * | 4.9281 ± 2.3704 | 12.4073 ± 3.3159 |
| High | 6 | 1.450 ± 1.329 * | 2.8525 ± 0.2716 ** | 5.8161 ± 2.3814 ** |
| Low | 6 | 2.580 ± 0.548 | 4.6217 ± 1.2715 | 5.0924 ± 1.8529 ** |

Table: Comparison of cytokine expression in the DSS group versus the Control group: ## $p < 0.01$. Comparison of the remaining groups with the DSS group: * $p < 0.05$, ** $p < 0.01$.

Table 2 reveals that, compared to the Con group, the DSS group showed significantly increased expression levels of miR-214, IL-6, and p-STAT3 ($p < 0.01$). Compared to the DSS group, the MS and High groups exhibited a significant decrease in miR-214 expression ($p < 0.05$). The High group showed a substantial decrease in IL-6 protein levels ($p < 0.01$), while the LP+*Ly* group exhibited a significant reduction in *p*-STAT3 levels ($p < 0.01$). This suggests that LP+*Ly* exerts its anti-inflammatory effects by modulating cytokine homeostasis.

### 3.3. LP+Ly and the Structure of the Gut Microbiota

3.3.1. Analysis of Intestinal Flora Diversity in Mice Following LP+*Ly* Intervention

The diversity of gut microbiota in mice after LP+*Ly* intervention is depicted in Figure 2.

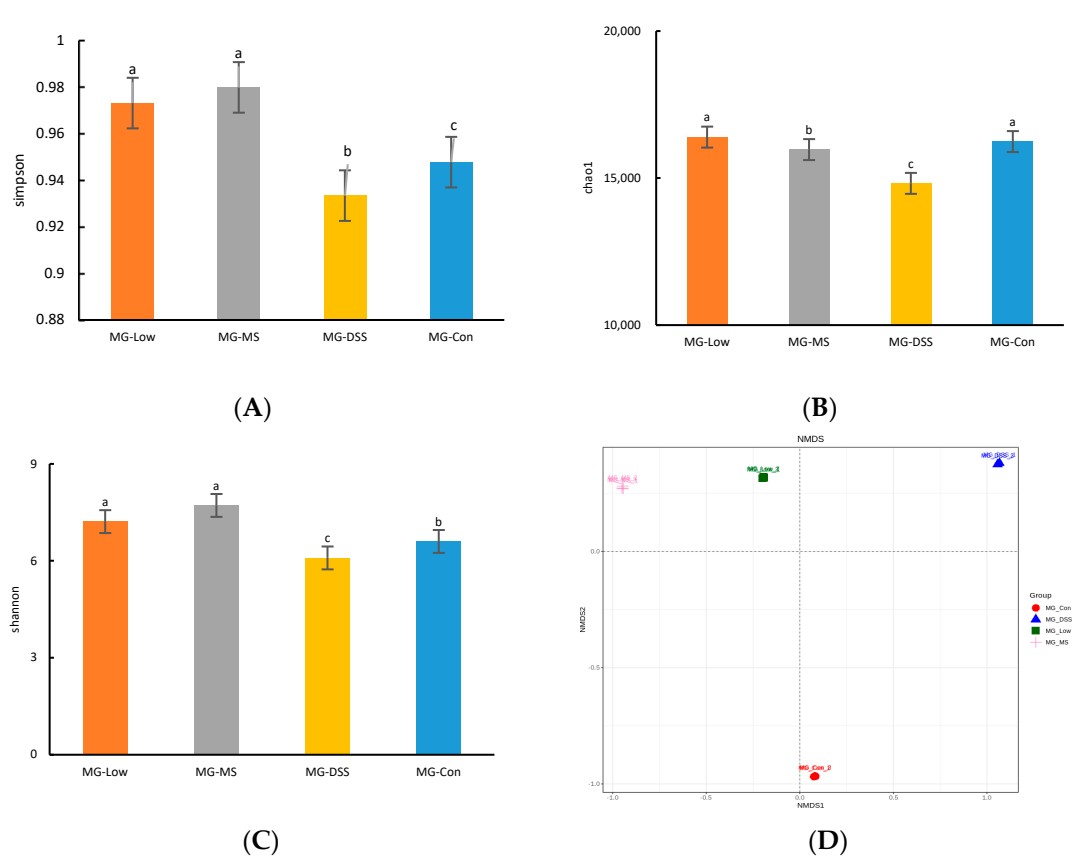

**Figure 2.** Diversity analysis of mice in each group. (**A**) Simpson index. (**B**) Chao index. (**C**) Shannon index. (**D**) Two−dimensional ordination plot of species NMDS analysis.

Figure 2 demonstrates that the α-diversity index in the DSS group is significantly lower compared to the other groups ($p < 0.05$). However, treatment with the medication and LP+*Ly* intervention significantly counteracts this decrease. The nonmetric multidimensional scaling (NMDS) analysis, which is based on the distance matrix of samples, points to substantial differences in β-diversity among the gut microbiota of mice from different groups (Figure 2D). The gut microbiota of UC mice exhibits distinct clustering and separation compared to the Con group. However, the Low group displays excellent within-group reproducibility and shows significant separation from the other groups. Moreover, the Low group is closest to the Con group, indicating the least difference and highest similarity in gut microbiota composition between these two samples.

### 3.3.2. Impact of LP+*Ly* on the Structure of Gut Microbiota

To explore the mechanism through which LP+*Ly* alleviates UC in mice, we first analyzed the effect of LP+*Ly* on the species composition of mice gut microbiota, as presented in Figure 3.

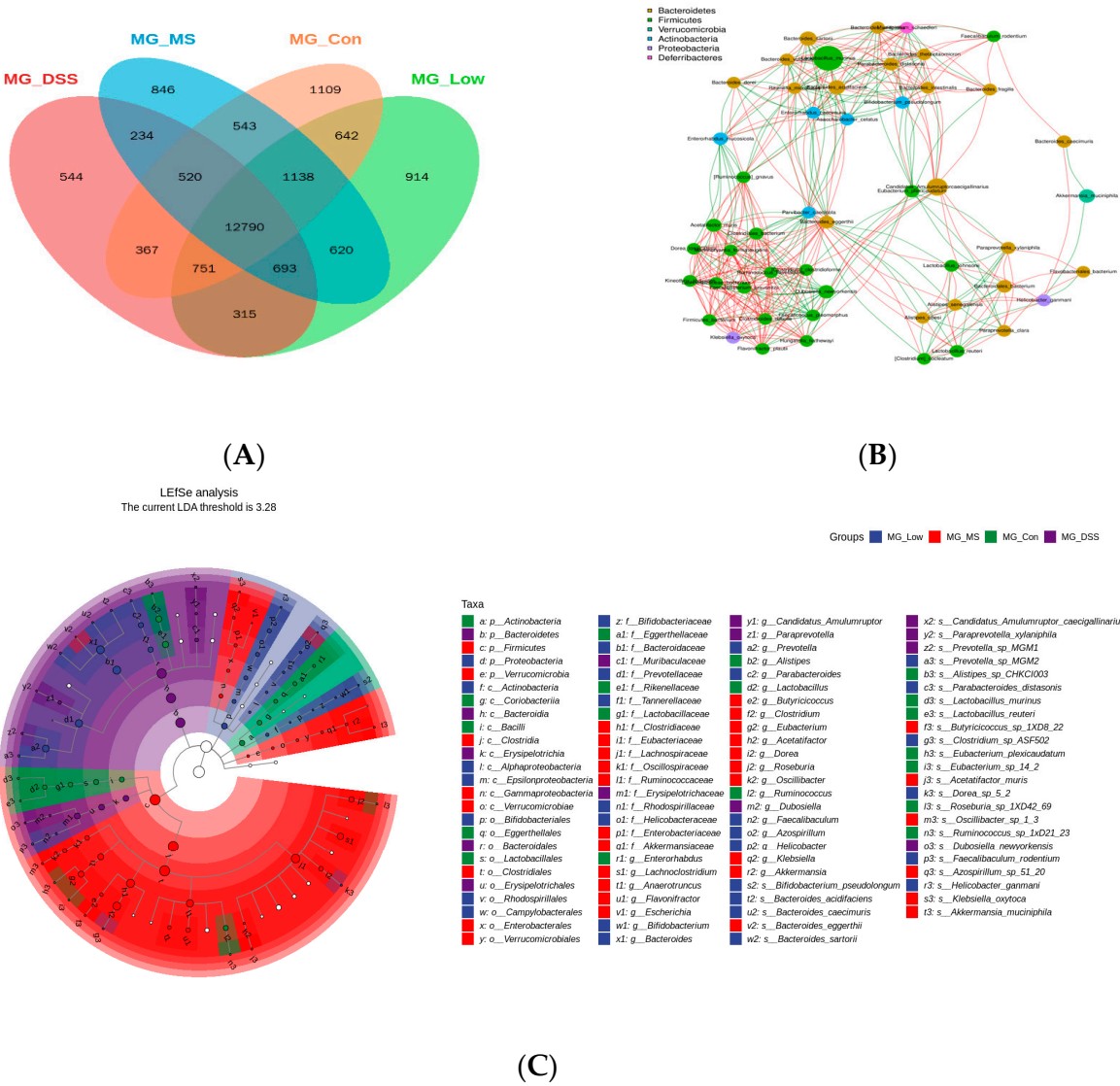

**Figure 3.** Effects of LP+*Ly* intervention on gut microbiota. (**A**) Venn diagram of species. (**B**) Network diagram of dominant microbial taxa at the species level (top 50 abundant species). (**C**) Clustered evolutionary tree analysis using LEfSe.

As shown in Figure 3A, at the species level, a total of 12,790 microbial species are shared across all groups. The DSS group possesses 544 unique species, the Con group has 1109 unique species, the Low group has 914 unique species, and the MS group has 846 unique species. Figure 3B indicates that dominant species in the *phylum Firmicutes*, including *Lactiplantibacillus murinus*, *Faecalibaculum rodentium*, *Firmicutes bacterium*, and *Lachnospiraceae bacterium*, exhibit close relationships and positive correlations with other species. Conversely, dominant species in the *phylum Actinobacteria*, such as *Enterorhabdus mucosicola*, *Enterorhabdus caecimuris*, *Asaccharobacter celatus*, and *Parvibacter caecicola*, demonstrate negative correlations with other species. LEfSe analysis results, from phylum to species, reveal the enrichment of specific taxa at varying taxonomic levels (Figure 3C). In the Con group, there is enrichment in *Lactobacillus*, *Lactobacillus ruminis*, and *Lactobacillaceae*. The DSS group exhibits enrichment in *Bacteroides acidifaciens*, *Akkermansia muciniphila*, and *Mucispirillum schaedleri* at the genus and species levels. In the MS group, there is enrichment in *Eggerthella* and *Akkermansia municiphila*. The Low group displays enrichment in *Eubacterium rectale*, *Anaerostipes*, and *Prevotella* at the genus level. Additionally, *Turicibacter*, *Bacteroides*, *Holdemania*, and *Dysgonomonas* have been identified as positively associated with colitis-related inflammation [12–15].

### 3.3.3. Analysis of the Effect of LP+*Ly* on the Functional Level of Intestinal Flora in Mice

Figure 4 shows the functional levels of the intestinal flora in each mouse group following the LP+*Ly* intervention.

According to Figure 4A, the Low group is in closer proximity to the Con group than to the DSS group. This suggests that the functional composition differences between the Low and Con groups are minimal. Figure 4B reveals a substantial difference between the DSS and Con groups, while the Low and MS groups are more similar to the Con group. The Low group clusters together with the Con group at the same hierarchical level, indicating minimal differences and the highest similarity between them. This further illustrates that the composition of the intestinal microbiota in the DSS group is significantly altered, but intervention with the medication and LP+*Ly* effectively mitigates these abnormal changes, making them more similar to the Con group. The protein gene KEGG enrichment analysis demonstrates that most proteins are primarily enriched in the categories of metabolism and genetic information processing at the first level of KEGG classification. Dominant functional categories include carbohydrate metabolism, amino acid metabolism, and nucleotide metabolism (Figure 4C).

### 3.4. Impact of LP+Ly on the Metabolome of Serum Samples

#### 3.4.1. Data Quality Control (QC) Analysis

The total ion chromatogram (TIC) of the QC samples is displayed in Figure 5 (the negative [neg] ion spectrum of QC is shown in Figure A1).

Figure 5 suggests that the intensities and retention times of chromatographic peaks in each sample are largely consistent, indicating minimal variation due to instrumental errors throughout the course of the experiment.

#### 3.4.2. Summary of Plasma Metabolite Identification Results

The identified metabolites from both positive (pos) and negative ion modes were consolidated, as depicted in Figure 6.

Figure 6 demonstrates that a total of 1123 metabolites were identified, comprising 631 metabolites in the pos mode and 492 metabolites in the neg mode. Based on the classification into chemical groups, 293 metabolites were categorized as lipids and lipid-like molecules, 227 metabolites as organic acids and derivatives, 126 metabolites as benzenoids, and 45 metabolites as phenylpropanoids and polyketides, among others.

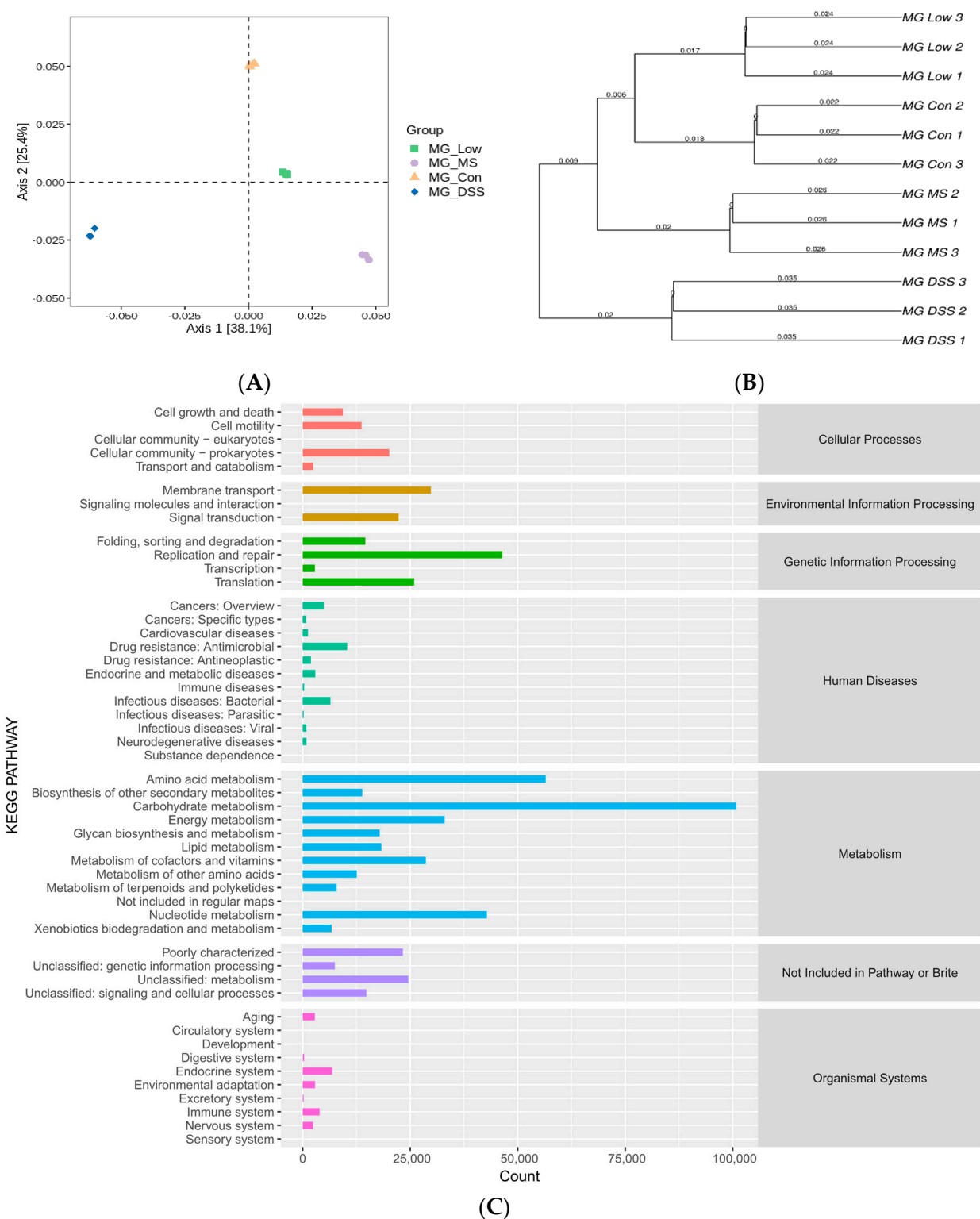

**Figure 4.** Functional analysis of the gut microbiota. (**A**) PCoA analysis showing two dimensional ordination of samples. (**B**) UPGMA cluster analysis. (**C**) KEGG metabolic pathway analysis.

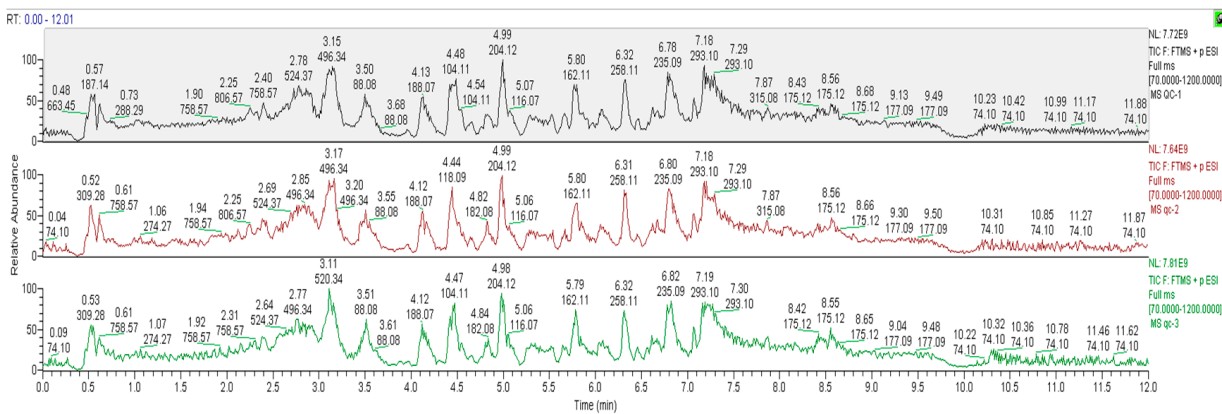

**Figure 5.** Overlay plot of total ion chromatograms (TIC) for positive ion QC samples (pos).

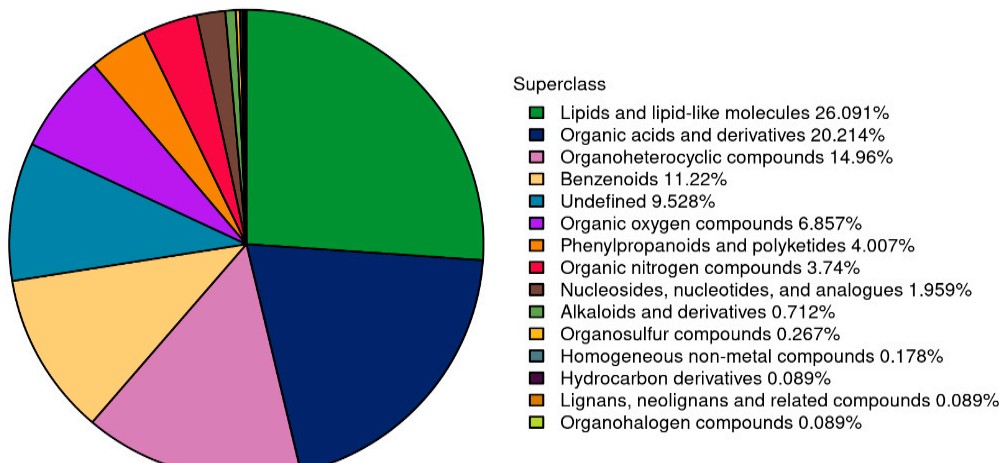

**Figure 6.** Proportion of identified metabolites in each chemical category.

3.4.3. Analysis of Differences between Groups of Plasma Metabolites and Permutation Test

The results regarding differential metabolites can be observed in Figure 7 (neg results in Figure A2).

According to Figure 7A,B, a total of 176 metabolites exhibited significant changes in expression between the Con and DSS groups, with 129 metabolites upregulated and 47 metabolites downregulated. There were 63 differentially expressed metabolites between the DSS and Low groups. Further, by comparing the overlapping differentially expressed metabolites between Con groups vs. DSS groups and DSS groups vs. Low group comparisons, a total of 63 metabolites were identified, 17 of which showed consistent expression patterns in the Con and Low groups. These 17 differentially expressed metabolites can be considered potential target metabolites involved in the alleviation of UC in mice by LP+*Ly* intervention.

According to Figure 7C,D, there is significant separation of metabolites between the DSS and Con groups, as well as between the DSS and Low groups. The permutation tests in Figure 7E,F indicate that as permutation retention decreases, both the $R^2$ and $Q^2$ of the random model gradually decrease, suggesting that the original model is not overfitted and exhibits robustness.

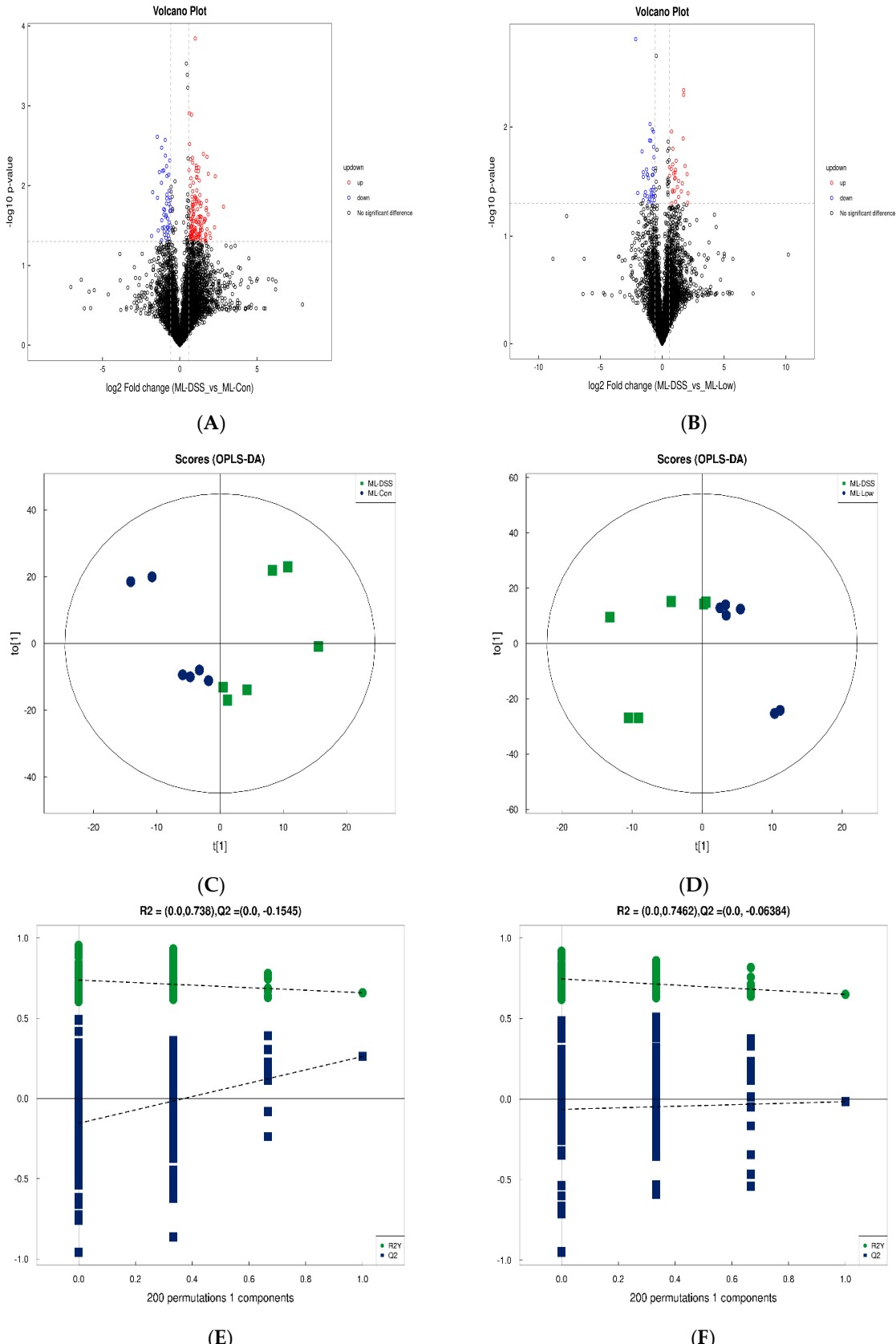

**Figure 7.** Presents the volcano plots and OPLS−DA analysis in pos. (**A**) DSS vs. Con; (**B**) DSS vs. Low; (**C**) DSS vs. Con; (**D**) DSS vs. Low; (**E**) shows the permutation test results for the comparison between DSS and Con groups, and panel (**F**) represents the permutation test results for the comparison between DSS and Low groups.

### 3.4.4. Screening of Differential Metabolites

By applying the selection criteria, significantly different metabolites between groups were screened, and the results are presented in Table 3 (neg results are shown in Table A3).

**Table 3.** Pos and neg DSS vs. Con significant difference metabolites.

| rt (s) | Name | VIP | FC | *p*-Value | *m/z* | Adduct |
|---|---|---|---|---|---|---|
| 476.16 | Fenarimol | 5.74 | 0.44 | 0.009 | 331.05 | [M+H]+ |
| 407.39 | L-citrulline | 2.34 | 1.46 | 0.033 | 198.08 | [M+Na]+ |
| 447.33 | Asp-Thr | 19.54 | 0.74 | 0.033 | 235.09 | [M+H]+ |
| 86.17 | Nudifloramide | 1.07 | 0.72 | 0.035 | 153.07 | [M+H]+ |
| 23.96 | Bergaptol | 6.86 | 2.44 | 0.006 | 201.02 | [M-H]− |
| 24.08 | 2-phosphonoethylphosphonic acid | 6.14 | 3.06 | 0.013 | 188.99 | [M-H]− |
| 331.83 | D-proline | 1.52 | 1.95 | 0.022 | 114.06 | [M-H]− |
| 23.94 | Pyrocatechol | 1.43 | 2.60 | 0.029 | 109.03 | [M-H]− |
| 25.02 | Daidzein 4′-sulfate | 1.03 | 4.61 | 0.030 | 333.01 | [M-H]− |
| 37.76 | 14-hydroxy-4z,7z,10z,12e,16z,19z-docosahexaenoic acid | 1.13 | 0.47 | 0.046 | 343.23 | [M-H]− |

In the comparison between the DSS and Con groups, a total of 10 significantly different metabolites were identified. In the pos mode, the upregulated differentially expressed metabolite was classified as organic acids and derivatives, specifically L-citrulline. The downregulated differentially expressed metabolites included Fenarimol from the class of benzenoids, Asp-Thr from the class of organic acids and derivatives, and nudifloramide from the class of organoheterocyclic compounds. In the comparison between the DSS and Low groups, a total of five significantly different metabolites were identified. In the pos mode, the upregulated differentially expressed metabolite belonged to the class of organic acids and derivatives, specifically L-citrulline. The downregulated differentially expressed metabolites included Fenarimol from the class of benzenoids and 2-ketohexanoic acid from the class of organic acids and derivatives.

### 3.4.5. Cluster Analysis of Differential Metabolites

To visually represent the expression pattern differences of significant differentially expressed metabolites across various samples, a clustering heatmap was generated, as shown in Figure 8 (neg heatmap in Figure A3).

Figure 8 reveals that, in the pos mode, Epoxiconazole, Feruloyl tyramine, Tenofovir, and others were significantly upregulated in the DSS group, while D-xylose was downregulated. However, the intervention with the drug and LP+*Ly* significantly reversed these trends. In the neg mode, phenyllactic acid, cytidine, and dodecanedioic acid were upregulated in the DSS group, but the intervention with the drug and LP+*Ly* helped normalize their levels toward those in the Con group. Consequently, carbohydrate metabolism and amino acid metabolism have been identified as the target metabolic pathways involved in alleviating UC in this study.

### 3.4.6. Analysis of the KEGG Metabolic Pathway with Differential Metabolites

The KEGG pathway annotations of the differentially expressed metabolites identified in both pos and neg ion modes are presented in Figure 9.

As depicted in Figure 9, for the DSS group vs. group Con comparison, the significantly upregulated metabolite pyrocatechol is associated with Metabolic pathways. The metabolite L-citrulline is involved in Arginine biosynthesis, Metabolic pathways, and the Biosynthesis of amino acids. D-proline participates in Arginine and proline metabolism and Metabolic pathways. The significantly downregulated metabolite nudifloramide is involved in Nicotinate and nicotinamide metabolism as well as Metabolic pathways. In the DSS group vs. Low group comparison, the significantly downregulated metabolite L-ascorbic acid in the Low group participates in Ascorbate and aldarate metabolism, Glutathione metabolism,

Metabolic pathways, and Vitamin digestion and absorption. L-citrulline is engaged in Arginine biosynthesis, Metabolic pathways, and the Biosynthesis of amino acids.

The graphical representation illustrating the pathogenesis of UC in mice and the mechanism of relief provided by LP+*Ly* at the metabolic level is shown in Figure 10.

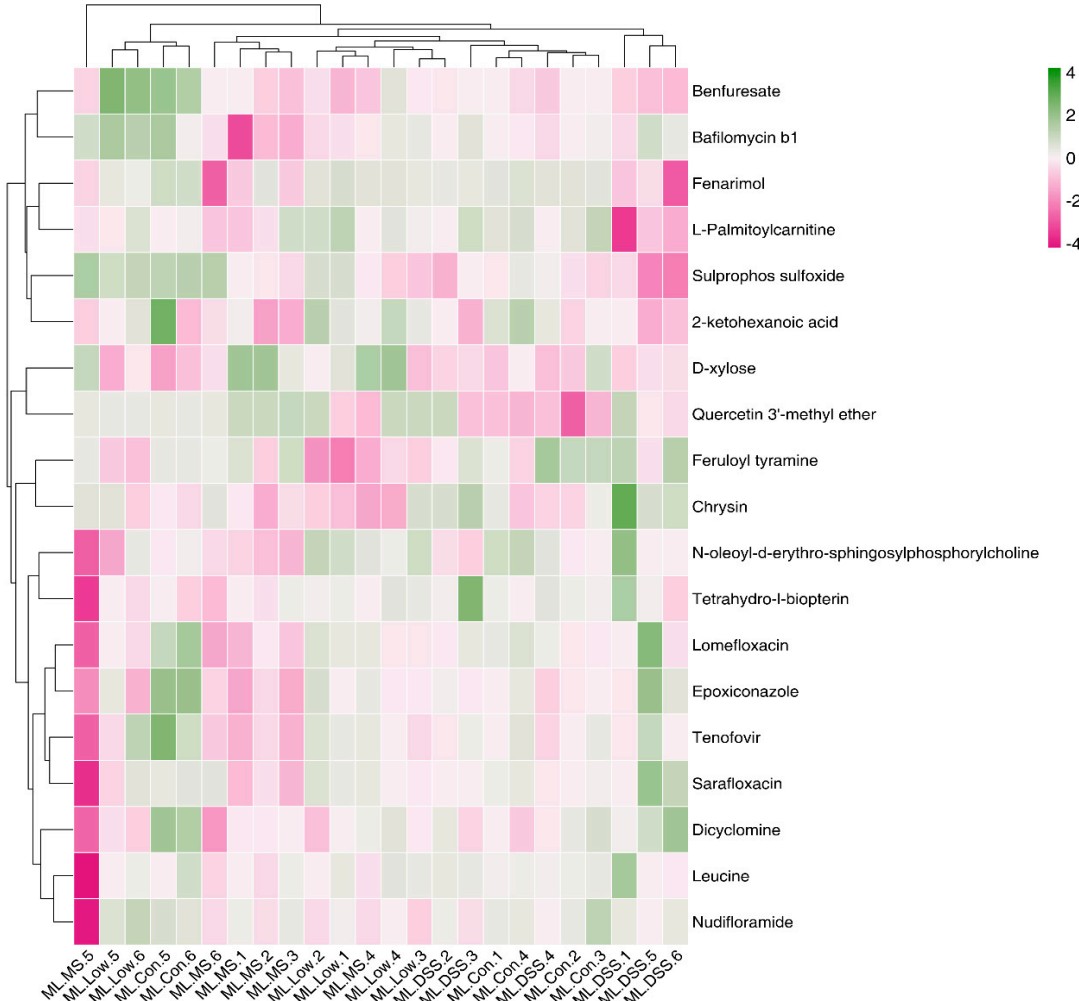

**Figure 8.** Heatmap analysis of significantly different metabolites (Pos).

According to the KEGG enrichment analysis in Figure 10, the comparison of the DSS group vs. the Con group shows that six significantly different metabolites map to five Metabolic pathways, including Arginine biosynthesis, Nicotinate and nicotinamide metabolism, and Arginine and proline metabolism, with Arginine biosynthesis being significantly enriched. The enrichment analysis for the DSS group vs. Low group indicates that five significantly different metabolites map to 13 metabolic pathways, including the Biosynthesis of amino acids, Citrate cycle, Arginine biosynthesis, Glucagon signaling pathway, Glutathione metabolism, and Vitamin digestion and absorption, with the first eight pathways being significantly enriched. Thus, it is inferred that these nine metabolic pathways are potential target pathways for the alleviation of UC in mice through LP+*Ly* intervention. Among them, Arginine biosynthesis, in which the metabolite L-citrulline is involved, exhibits the strongest correlation, and the intake of arginine is vital for suppressing inflammation and carcinogenesis [16]. Please refer to Figure 11 for specific details on metabolic pathways.

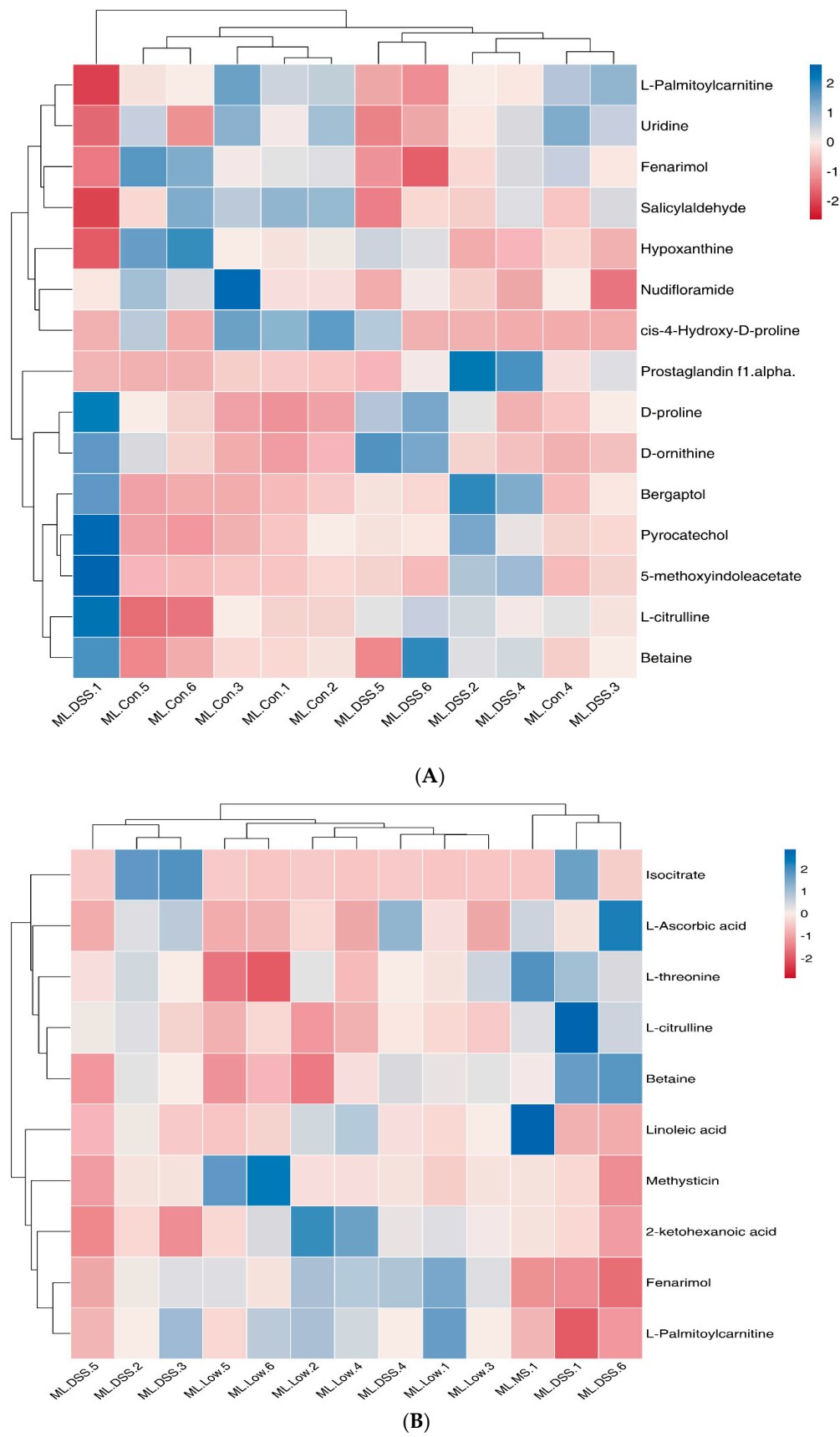

**Figure 9.** Heatmap of differential metabolite clustering in the KEGG pathway. (**A**) DSS vs. Con; (**B**) DSS vs. Low.

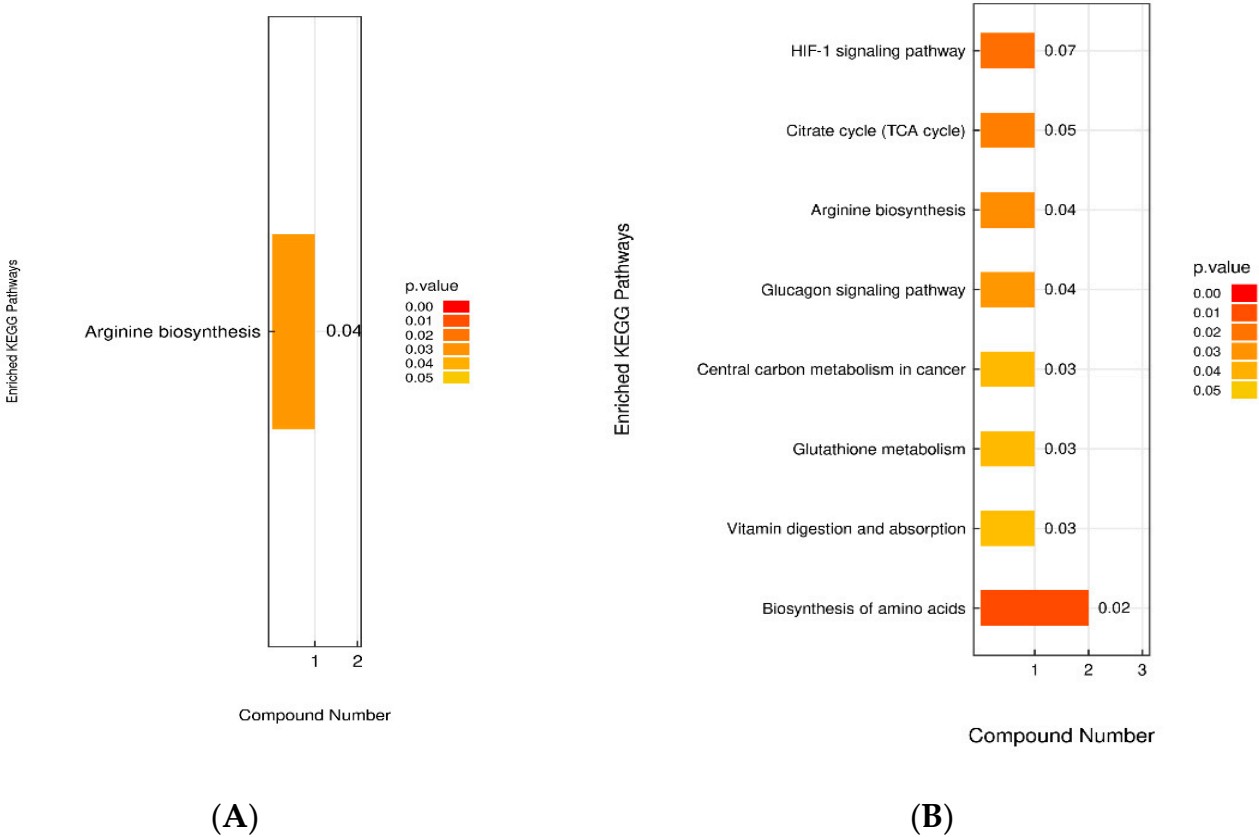

**(A)** **(B)**

**Figure 10.** KEGG pathway enrichment map (**A**): DSS vs. Con; (**B**): DSS vs. Low.

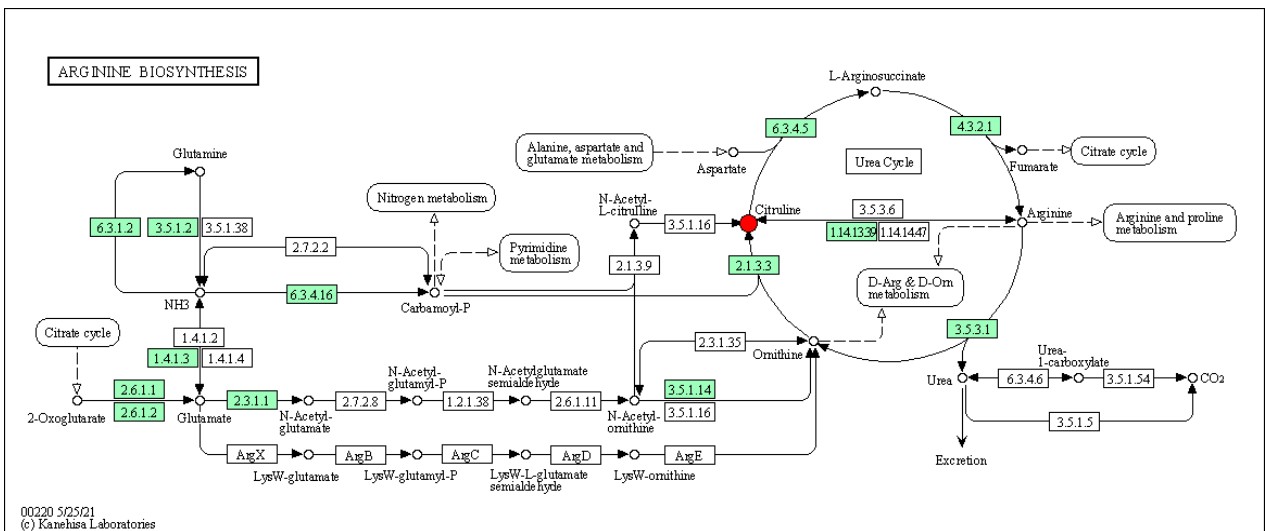

**Figure 11.** Arginine biosynthesis pathway map. In the metabolic pathway diagram, red circles indicate upregulated metabolites, and light green boxes represent species-specific proteins.

Figure 11 elucidates that in the urea cycle, the upregulation of the metabolite guanidinoacetic acid results in alterations in L-arginine succinate, subsequently influencing the biological metabolic pathways associated with arginine. This offers a preliminary insight into how LP+*Ly* exerts anti-inflammatory and immunomodulatory effects on UC by affecting relevant pathways of Arginine biosynthesis.

### 3.5. Combined Resonance Analysis of Intestinal Flora and Metabolites

To delve further into the restorative effects of LP+*Ly* intervention on UC in mice, a combined analysis of mouse gut microbiota and metabolites was performed. The canonical correspondence analysis (CCA) plots for both pos and neg ion modes are depicted in Figure 12.

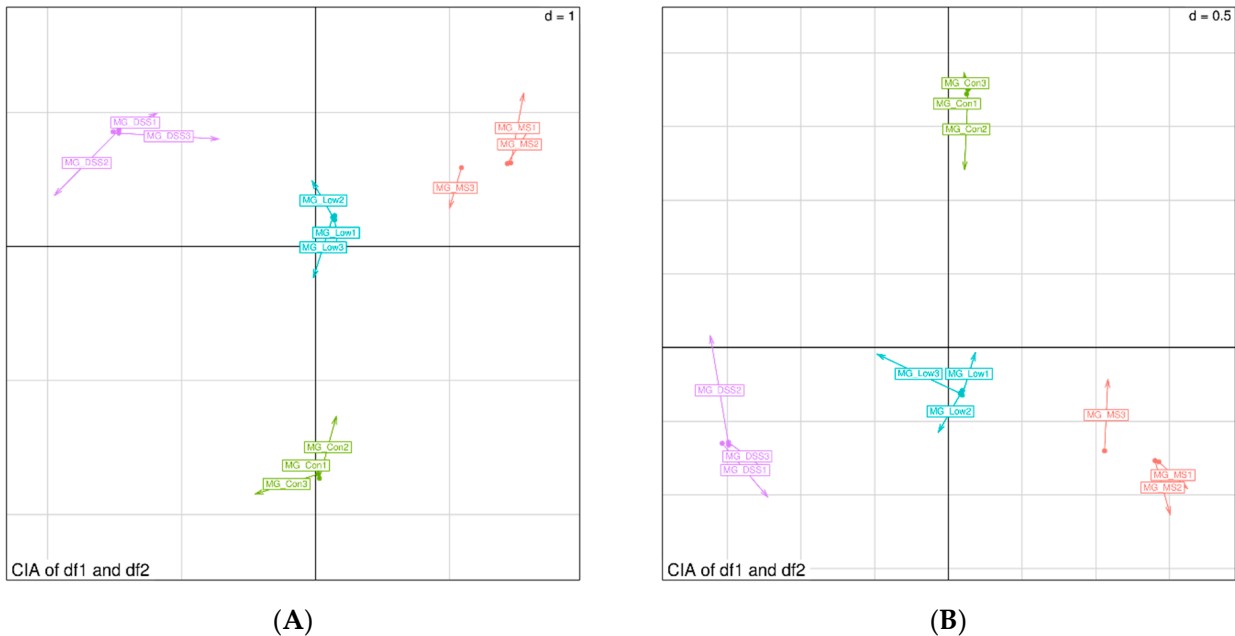

(**A**)                                                                        (**B**)

**Figure 12.** CIA two-dimensional alignment chart (**A**) pos; (**B**) neg.

Figure 12 reveals that, compared to the Con group, the arrow length is longest for the DSS group, indicating significant discrepancies between the gut microbiota composition and metabolomics data in the DSS group. Conversely, the MS and Low groups exhibit patterns closer to the Con group, with the Low group demonstrating greater similarity in pos ion mode. This suggests that through drug and LP+*Ly* intervention, the mouse gut tends to return to a healthier state.

A heatmap illustrating the correlations between microbial species and metabolite detection results is presented in Figure 13 (neg heatmap in Figure A4).

As per Figure 13, there is a close interrelationship between plasma metabolites and microbial genera. There are positive correlations of *1-palmitoyl-2-docosahexaenoyl-sn-glycero-3-phosphocholine* with *Odoribacter* and *Helicobacter*, *Heptadecasphinganine* with *Paraprevotella* and *Alloprevotella*, *Prostaglandin E1 alcohol* with *Barnesiella*, and *Leucine* with *Prevotella* and *Faecalibaculum*, among others. Conversely, there are negative correlations between *Betaine* with *Odoribacter*, *Arachidonoylthiophosphorylcholine* with *Akkermansia* and *Desulfovibrio*, *Leucine* with *Akkermansia*, and *L-hydroxyarginine* with *Alistipes* and *Parvibacter*, among others. These findings suggest that LP+*Ly* regulates plasma metabolism by modulating the gut microbiota, thereby influencing the systemic inflammatory response.

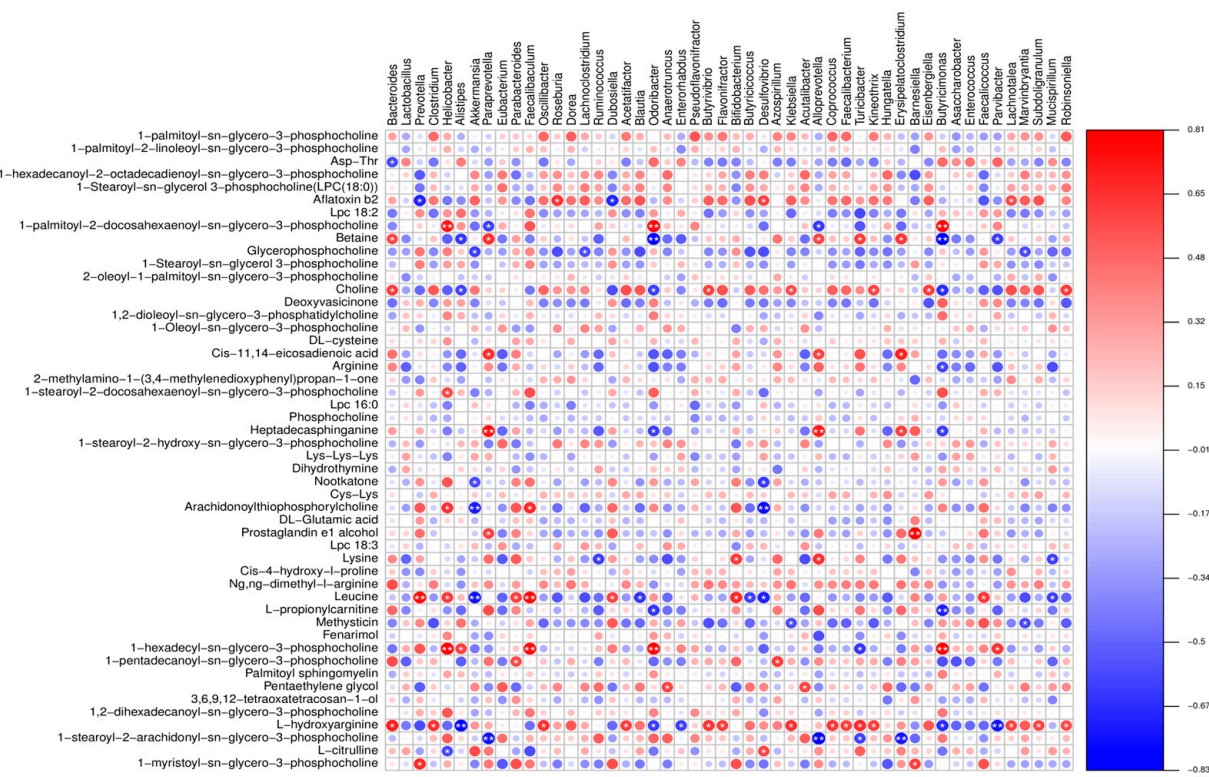

**Figure 13.** Spearman rank association heatmap between intestinal flora and potential metabolites (pos).

## 4. Analysis and Discussion

Plant-based diets have been associated with protective effects against various diseases by reducing oxidative stress, modulating cytokine secretion, and regulating gut microbiota to fortify the intestinal barrier [17]. In this experiment, we employed microbial fermentation to ferment goji berries with the goal of producing beneficial microorganisms and secondary metabolites. This process was expected to enhance the efficacy of LP+*Ly* in alleviating UC.

One of the inflammatory factors, P-STAT3, is indicative of inflammation, where a higher degree of phosphorylation suggests increased inflammation. IL-6, a pro-inflammatory cytokine, can stimulate the phosphorylation of P-STAT3 [18]. Additionally, dysbiosis of the gut microbiota due to defects in immune system-related genes can cause an imbalance in the microbial community, consistent with existing studies [19]. Farraye et al. [20] also established a significant positive correlation between the expression of the miR-214 gene and the severity of UC in patients.

In our study, through immunohistochemical analysis, we measured the levels of P-STAT3, IL-6, and miR-214 in colonic tissues. Our findings showed that LP+*Ly* effectively suppressed the mRNA expression of pro-inflammatory cytokines, such as P-STAT3 and IL-6. Consequently, the suppression of pro-inflammatory cytokines by LP+*Ly* may be essential for mitigating DSS-induced intestinal inflammation and damage.

DSS-induced UC in mice is typified by symptoms such as weight loss, bloody diarrhea, colon shortening, and intestinal damage [21,22]. Moreover, UC patients often exhibit a decline in the number of goblet cells in the colon, leading to increased mucosal barrier permeability [23]. Our study found that LP+*Ly* indeed mitigated these pathological symptoms and ameliorated histological damage in UC mice. Nonetheless, LP+*Ly* comprises a multifarious mixture of active components. It is therefore plausible that its effectiveness against UC is attributable to the synergistic action of multiple constituents. Further studies are necessary to determine how these components interact to alleviate UC and to identify the primary active components.

The gut microbiota is implicated in various physiological functions, such as regulating inflammatory responses, controlling appetite, and maintaining intestinal integrity [24]. Disruption of the gut microbiota, termed dysbiosis, can serve as an important marker of the pathological and physiological state of the host. Our study demonstrated that gut microbiota diversity and richness diminished in the DSS-induced group of mice, as evidenced by a decrease in the species richness, Chao-1 index, Shannon index, and Simpson index. However, the administration of medication and LP+*Ly* reversed these trends, with LP+*Ly* nearly restoring these indicators to levels observed in healthy mice.

Additionally, LP+*Ly* altered the composition of the gut microbiota in UC mice, including changes in the abundance of specific bacterial phyla such as *Bacteroidetes* and *Firmicutes*. This could be attributed to the polysaccharides present in LP+*Ly*, as polysaccharides are known to be effective in modulating the gut microbiota. In this study, the occurrence of UC significantly reduced the relative abundance of *Lactiplantibacillus*. Unfortunately, intervention with LP+*Ly* did not reverse this decreasing trend. Additionally, LP+*Ly* increased the relative abundance of *Helicobacter*. It is worth noting that *Helicobacter* has been negatively associated with IBD [25], and its depletion can disrupt the balance of the gut microbiota [26]. This is consistent with the significant decrease in relative abundance of *Helicobacter* seen in the DSS group. At the genus level, the gut microbiota structure was disrupted in the DSS group, with a decreased relative abundance of beneficial bacteria (such as *Lactiplantibacillus*, *Clostridium*, and *Akkermansia*), while administration of LP+*Ly* reduced the relative abundance of harmful bacteria (such as *Candidatus Amulumruptor*, *Odoribacter*, and *Alloprevotella*), indicating a broad-spectrum antibacterial effect of LP+*Ly*. Moreover, in this experiment, the relative abundance of short-chain fatty acid (SCFA)-producing bacteria, such as *Alistipes*, *Clostridium*, and *Roseburia*, was decreased in the DSS group. Conversely, the relative abundance of SCFA-producing bacteria, including *Bacteroides* and *Parabacteroides*, was higher in the groups treated with the drug and LP+*Ly* compared to the Con group. This indicates that the alleviating effect of LP+*Ly* on UC mice partially originates from its modulation of the gut microbiota.

LP+*Ly* also influenced the relative abundance of SCFA-producing bacteria, which are crucial for the proliferation of intestinal epithelial cells and the maintenance of epithelial barrier function [27–29]. The abundance of such bacteria is reduced in IBD patients, whereas probiotic bacteria that produce SCFAs have been found to alleviate clinical symptoms in these patients [30,31].

In recent times, research has highlighted the significant role of dietary amino acids in preventing and treating intestinal inflammation. Amino acids are implicated in protecting the intestine through mechanisms such as inhibiting intestinal epithelial cell apoptosis, alleviating intestinal inflammation, reducing oxidative stress reactions, and restoring intestinal barrier function [32,33]. Our study employed metabolomic analysis to investigate the plasma metabolites of UC mice post-intervention with medication and LP+*Ly*. The goal was to elucidate the metabolic mechanisms through which LP+*Ly* alleviates UC. Analysis of selected metabolites revealed distinct trends in small molecule metabolites such as fenarimol, L-citrulline, and 2-ketohexanoic acid during DSS treatment. These trends indicated significant differences in the relative abundance of metabolites among the groups. UC mice displayed significant metabolic disturbances, which LP+*Ly* was able to partially normalize.

Notably, the NO in the inflamed bowel is generated mostly by an inducible nitric oxide synthase (NOS2) from L-arginine (further referred to as arginine). L-citrulline, an intermediate metabolic amino acid produced by small intestinal cells, was modulated by LP+*Ly*. L-citrulline is involved in the urea cycle, arginine synthesis, and nitric oxide production. Previous research has shown that certain amino acids, such as glutamine and arginine, can influence the pathogenesis of IBD [34]. In DSS-treated mice, L-arginine supplementation reduced colonic inflammation [35–37]. Additionally, L-arginine treatment of DSS-treated mice resulted in increased ex vivo migration of colonic epithelial cells, suggesting an increased capacity for wound repair [38]. Moreover, plasma guanidinoacetate levels are considered promising quantitative biomarker for diagnosing gastrointestinal

complications [39]. Our study observed that plasma guanidinoacetate levels in UC mice were significantly higher compared to the Con group. Changes in guanidinoacetate levels are associated with arginine biosynthesis in the body. LP+*Ly* intervention resulted in changes in L-citrulline levels, which in turn regulated arginine biosynthesis, contributing to the alleviating effects of LP+*Ly*. KEGG enrichment analysis suggested that the arginine biosynthesis pathway, in which L-citrulline participates, may be a potential therapeutic target for the action of LP+*Ly*.

Furthermore, our correlation analysis showed a strong association between numerous gut microbiota and differential metabolites. Previous studies have shown that supplementation with arginine can restore gut microbiota diversity [36]. Therefore, we postulate that LP+*Ly* exerts its beneficial effects by modulating the gut microbiota and influencing amino acid metabolism.

In summary, LP+*Ly* exhibited promising anti-inflammatory effects in DSS-induced UC in mice. This study provides scientific evidence supporting the potential clinical application of LP+*Ly* in the prevention and treatment of UC and may also contribute to the development of novel therapeutic approaches for UC.

## 5. Conclusions

In this study, we compared the effects of a standard drug and LP+*Ly*, fermented by *Lactiplantibacillus plantarum* NXU0011, on the alleviation of colitis. Our findings indicate that LP+*Ly* holds significant promise for mitigating colitis, demonstrating a more substantial impact compared to the standard drug. LP+*Ly* is instrumental in alleviating DSS-induced intestinal inflammation, largely due to its ability to modulate the gut microbiota and restore the disrupted microbial structure within the gut. Additionally, we observed that the protective effect of LP+*Ly* on plasma metabolites is intrinsically linked to its role in regulating the gut microbiota.

In essence, LP+*Ly* could be considered a promising probiotic product derived from L with the potential to improve intestinal inflammation. However, this study leaves some pertinent questions unanswered: The differences in gut microbiota and physiological status between mice and humans [40] make it uncertain as to whether the effects of LP+*Ly* in our study would translate similarly to human subjects with UC, and the long-term consumption of probiotic products, though potentially beneficial, could pose risks to the host. For instance, the extended administration of inulin has been associated with the induction of jaundice-type hepatocellular carcinoma in dysbiotic mice [41].

Therefore, future studies must prioritize investigation of the efficacy and safety of LP+*Ly* in a human physiological setting to ascertain its viability as a therapeutic intervention for UC.

**Author Contributions:** Conceptualization, M.N. and Q.J.; methodology, M.N. and Q.J.; software, G.G.; validation, M.N., Q.J. and G.G.; formal analysis, M.N. and L.P.; investigation, M.N., R.Z., H.Z. and L.P.; resources, M.N., Q.J. and L.P.; data curation, not applicable; writing—original draft preparation, M.N.; writing—review and editing, M.N., Q.J. and G.G.; visualization, L.P.; supervision, Q.J., Y.W., H.Z. and L.P.; project administration, Q.J., H.Z. and L.P.; funding acquisition, L.P. All authors have read and agreed to the published version of the manuscript.

**Funding:** This work was funded by the Key R&D Program of Ningxia (Grant No. 022004220014). Key R&D Programme Project (2020BBF02023); Autonomous Region Key R&D Programme (Talent Attraction Special Project) (2020BEB04014); Ningxia Natural Science Foundation (2021AAC03023).

**Institutional Review Board Statement:** The study was conducted in accordance with the Declaration of Helsinki, and approved by the Experimental Animal Ethics Committee of Sichuan Lilaisinuo Biological Technology Co., LTD. (Approval No: LLSN-2022014 Approval date: 3 June 2022).

**Data Availability Statement:** No new data were created or analyzed in this study. Data are contained within the article.

**Conflicts of Interest:** The authors declare that they have no known competing financial interests or personal relationships that could have appeared to influence the work reported in this paper.

## Appendix A

**Table A1.** Disease Activity Index Scoring Sheet.

| Scoring/Points | Weight Loss/% | Stool Characteristics | Rectal Bleeding (Occult Blood/Gross Blood in Stool) |
|---|---|---|---|
| 0 | 0 | Normal | Occult blood |
| 1 | 1–5 | Stool consistency: Soft but formed | Occult blood |
| 2 | 6–10 | Stool consistency: Soft | Visible blood traces |
| 3 | 11–18 | Stool consistency: Very soft and loose | Visible, significant blood |
| 4 | >18 | Watery diarrhea | Profuse bleeding |

**Table A2.** PCR Reaction System.

| Reagent | 20 μL System | Final Concentration |
|---|---|---|
| 2X SYBR Green Mix | 10 μL | 1× |
| RT Product | 2 μL | |
| Bulge-Loop$^{TM}$ miRNA Forward Primer (5M) | 0.8 μL | 200 nM |
| Bulge-Loop$^{TM}$ Reverse Primer (5M) | 0.8 μL | 200 nM |
| RNase-free $H_2O$ | To 20 μL | |

**Table A3.** Significantly different metabolites between the DSS group and the Low group in pos and neg.

| rt (s) | Name | VIP | FC | *p*-Value | *m/z* | Adduct |
|---|---|---|---|---|---|---|
| 510.32 | 2-ketohexanoic acid | 1.61 | 0.53 | 0.0133 | 131.08 | [M+H]+ |
| 407.39 | L-citrulline | 1.98 | 1.33 | 0.0332 | 198.08 | [M+Na]+ |
| 476.16 | Fenarimol | 4.00 | 0.55 | 0.0498 | 331.05 | [M+H]+ |
| 169.08 | L-Ascorbic acid | 2.10 | 5.44 | 0.021 | 197.01 | (M+Na-2H)− |
| 584.04 | Isocitrate | 3.28 | 33.75 | 0.038 | 173.01 | (M-$H_2O$-H)− |

## Appendix B

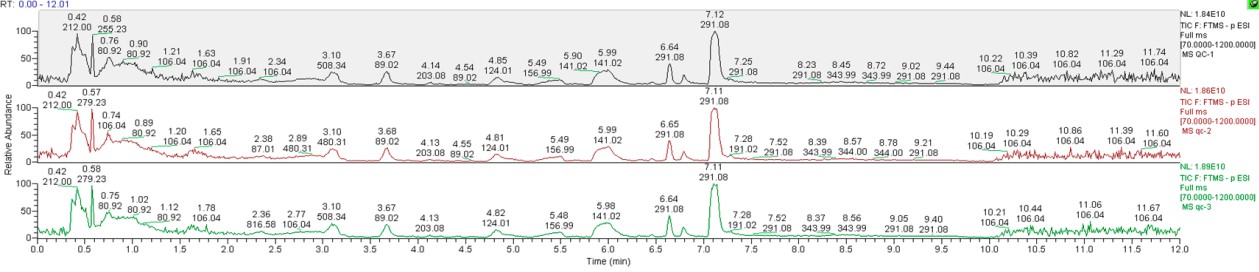

**Figure A1.** Overlay spectra of TIC for QC samples in neg.

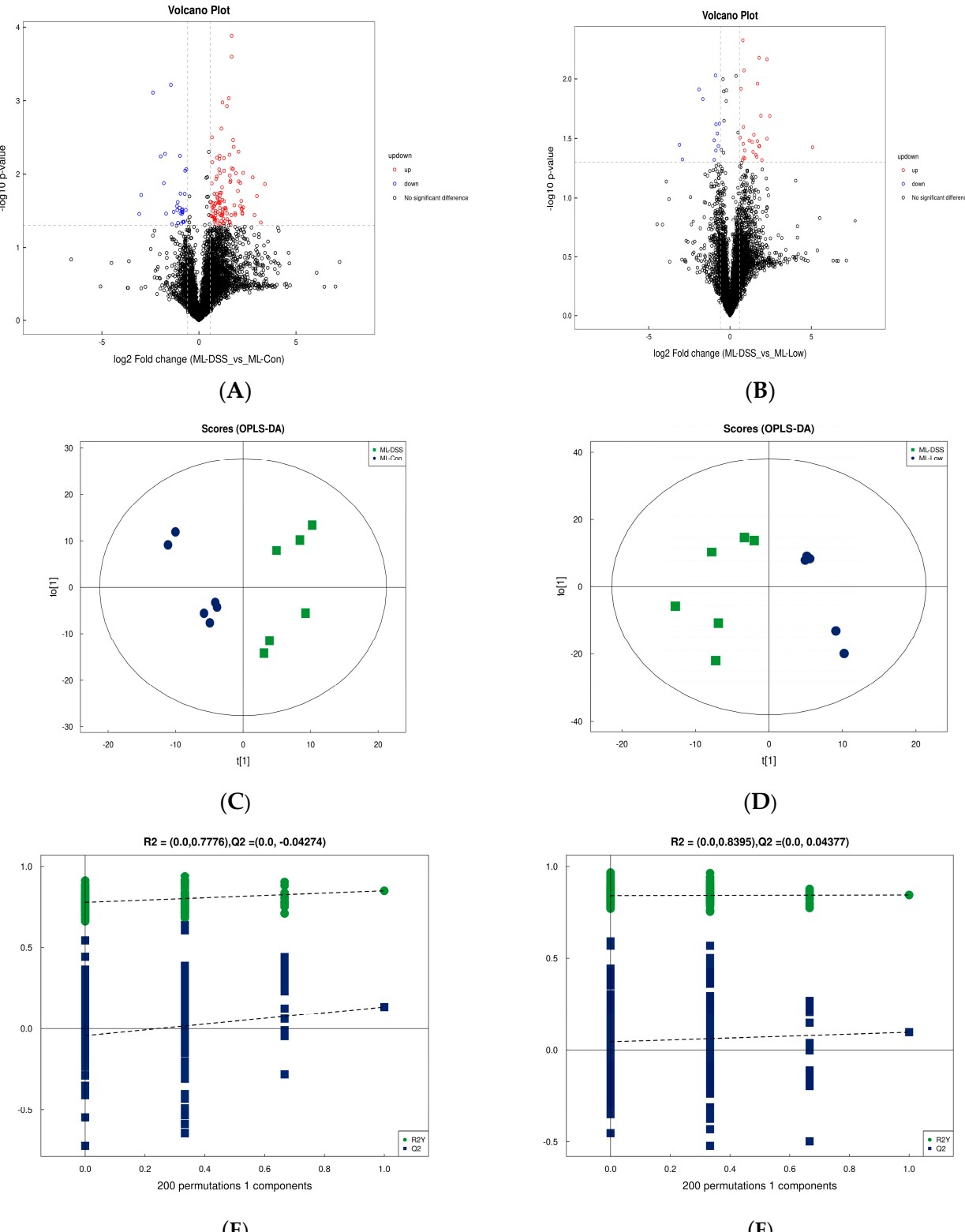

**Figure A2.** Volcano plot and OPLS−DA analysis in neg. (**A**) DSS vs. Con; (**B**) DSS vs. Low; (**C**) DSS vs. Con; (**D**) DSS vs. Low; (**E**) DSS vs. Con Permutation tes; (**F**) DSS vs. Low Permutation tes.

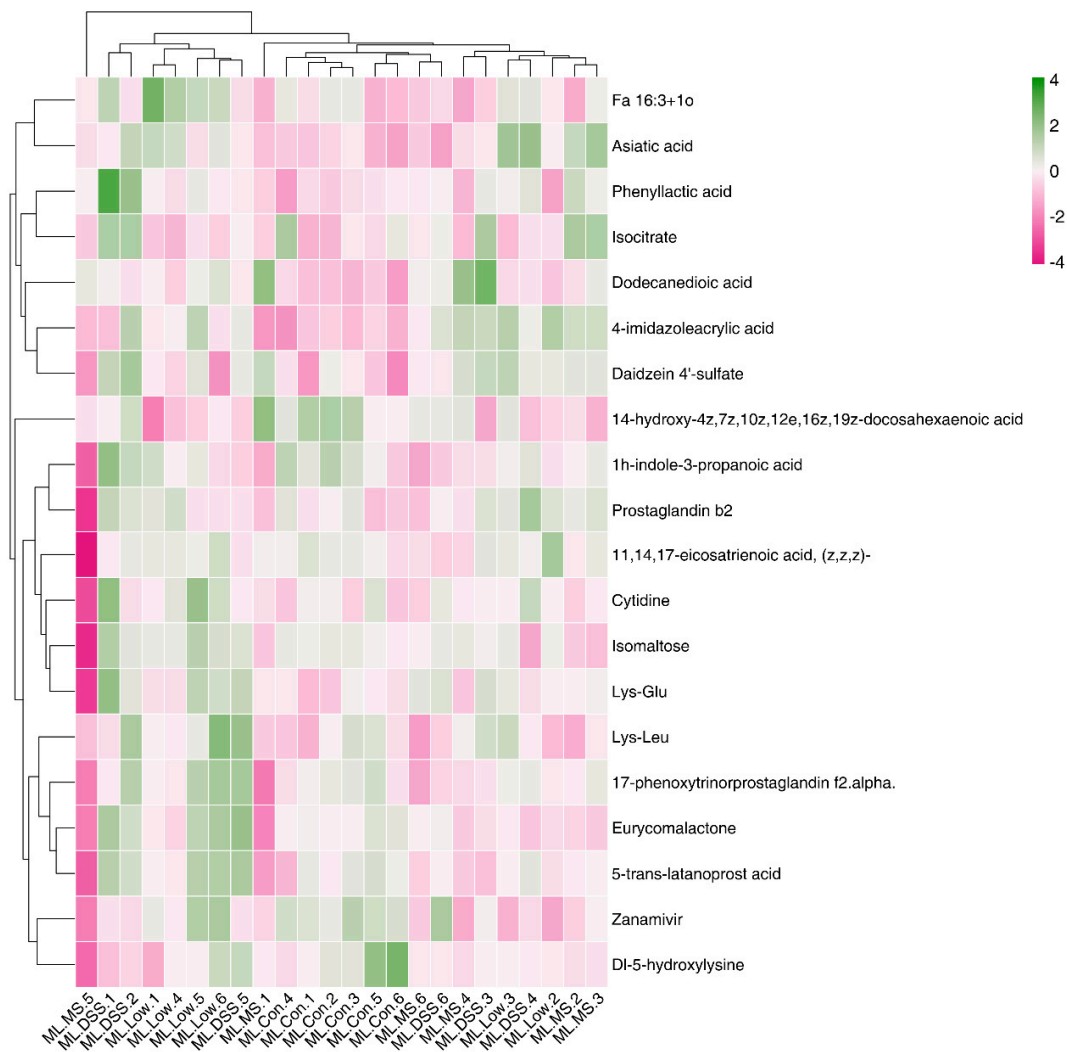

**Figure A3.** Hierarchical clustering heatmap of significantly different metabolites in neg.

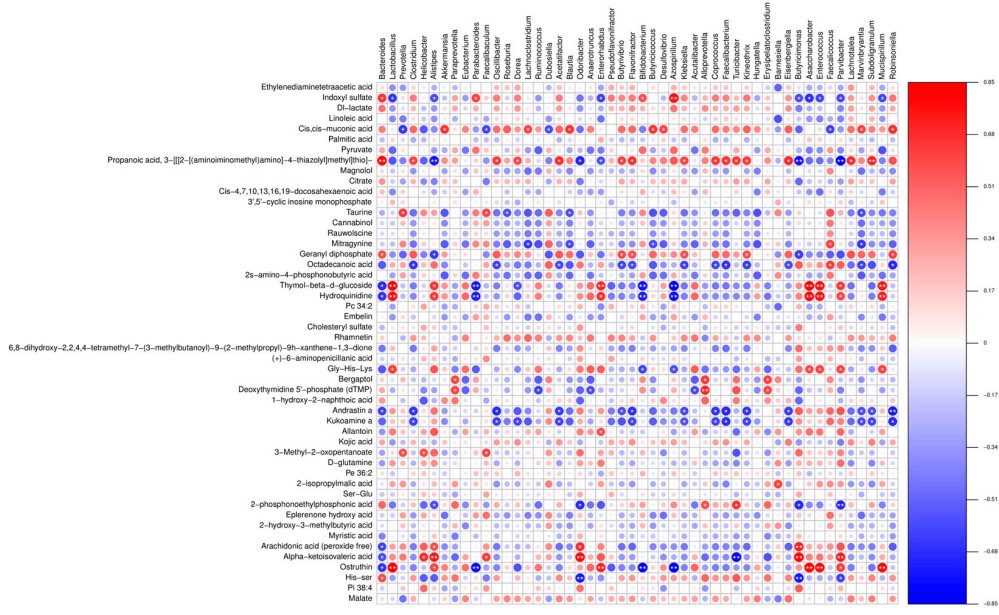

**Figure A4.** Spearman rank correlation heatmap between metabolomics data in neg and bacterial abundance.

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
