# Peer review of "Alleviating Effect of Lactiplantibacillus plantarum NXU0011 Fermented Wolfberry on Ulcerative Colitis in Mice"

_fermentation, doi:10.3390/fermentation9110971_

Round 1

Reviewer 1 Report

Comments and Suggestions for Authors

The manuscript describes an interesting study aimed at investigating the influence of fermented wolfberry on ulcerative colitis in mice. The design of the study and the methodology used are appropriate in my opinion. The results are interesting, although they have limitations that the authors mention. English is good. It is necessary to make some formal corrections that are detailed:

  - Update the taxonomy of Lactobacillus plantarum (title, abstract, etc.), as it appears in Conclusions

- The genus and species of microorganisms must always be written in italics. Review all text.

- line 22:"...and Akkermansia genus."

- lines 33 and 47: Lycium barbarum L. ? You could write "(L)"

- 2.1, lines 48-49: not understood, rewrite

- line 147: representative

- line 208: "Firmicutes", in italics

- line 209: "phylum", not in italics

- What is the origin of the L. plantarum strain used and why was this strain chosen to do the study?

Author Response

Thank you for your valuable suggestions and comments. We appreciate your thorough review of our manuscript. Below are our responses to each of your points:

**Comment 1:** Update the taxonomy of Lactobacillus plantarum (title, abstract, etc.), as it appears in Conclusions

**Response 1:** Thank you for pointing this out. We have updated the taxonomy of Lactobacillus plantarum throughout the manuscript, including the title, abstract, and Conclusions section, to align with the latest classification.

**Comment 2:** The genus and species of microorganisms must always be written in italics. Review all text..

**Response 2:** We appreciate your observation. We have carefully reviewed the entire manuscript and ensured that all genus and species names of microorganisms are written in italics as per the standard convention.

**Comment 3:** In line 22: "...and Akkermansia genus."

**Response 3:** Thank you for pointing out the error. We have made the necessary correction in line 22 to properly refer to the "Akkermansia" genus.

**Comment 4:** In lines 33 and 47: Lycium barbarum L.? You could write "(L)."

**Response 4:** We acknowledge your suggestion. The mention of "Lycium barbarum L." has been revised to "(L)" in lines 33 and 47 to adhere to the appropriate format.

**Comment 5:** In section 2.1, lines 48-49: Not understood, rewrite.

**Response 5:** Thank you for highlighting this issue. We have carefully rephrased the content in section 2.1, lines 48-49, to enhance clarity and improve understanding.

**Comment 6:** In line 147: "representative."

**Response 6:** We appreciate your feedback. The term "representative" in line 147 has been appropriately addressed and revised.

**Comment 7:** In line 208: "Firmicutes," in italics.

**Response 7:** Thank you for the correction. We have italicized "Firmicutes" in line 208 as per the specified format.

**Comment 8:** In line 209: "phylum," not in italics.

**Response 8:** We have corrected the formatting in line 209, removing the italics from the term "phylum" to comply with the appropriate style.

**Comment 9:** What is the origin of the L. plantarum strain used, and why was this strain chosen for the study?

**Response 9:** We evaluated 10 strains of Lactobacillus isolated from Ningxia river water for digestive stress tolerance, DPPH radical scavenging ability, antimicrobial activity, α-amylase and α-glucosidase inhibitory activity. The best performing strain selected from these tests was named Lactiplantibacillus plantarum NXU0011. It has a high survival rate in saliva, gastric fluid and intestinal fluid, and has the potential to be a good regulator of probiotic effects. We hypothesized that combining this probiotic with goji berries and obtaining a lyophilized bacterial powder with potential probiotic benefits through fermentation might alleviate the effects of colitis.

Please feel free to reach out if you have any further questions or require additional clarification regarding our responses. We once again appreciate your valuable feedback, which has significantly contributed to the enhancement of our research.

Reviewer 2 Report

Comments and Suggestions for Authors

This is an interesting manuscript on the effect of L. plantarum NXU0011 Fermented Lycium barbarum L. on Ulcerative Colitis in Mice. 

Overall, it is a well-written paper, highlighting some useful findings that can help the microbial fermentation field.

The methodology is sound and the data presentation is overall clear and informative.

Few minor corrections: 

1. In Abstract and Introduction, mention that Lycium barbarum L. is goji berry plant

2. In Abstract, include a concluding sentence about how this study will lead to new UC treatments.

3. The Introduction is too short, expand by mentioning similar studies with probiotics vs UC.

4. Introduction: explain how the tested cytokines are involved in UC pathogenesis and the role of arginine in UC.

5. Methods: Line 49, pasteurised not sterilised. Line 62, viable L. plantarum bacteria. Was this only bacteria or LP+Ly? Explain that mesalazine is used vs UC.

6. All species names must be in italics throughout the text.

7. Figure 2D is too blurry and imposible too read even when zoomed. Same for figure 3.

8. Figure 3 needs better description in the figure legend.Same for Figure 12.

9. Change Lactobacillus to Lactiplantibacillus throughout the text.

10. Discussion, line 469, explain how arginine affects UC. The mode of action here is not clear. 

11, Optional: if you had one control group with bacteria only and another with goji berry only, you could determine which component of LP-ly is more effective.

Comments on the Quality of English Language

Very good.

Author Response

For research article

Response to Reviewer X Comments

We appreciate Reviewer X's thoughtful feedback on our research article. We have carefully considered each point raised and made the following revisions in accordance with the suggestions. The corrected portions have been highlighted in the resubmitted document:

  1. **Abstract and Introduction:** We have included a statement in both the Abstract and Introduction clarifying that *Lycium barbarum L.* refers to the goji berry plant, providing clear context for readers.
  2. **Abstract Conclusion:** A concluding sentence has been added to the Abstract, highlighting the potential implications of our study for the development of new treatments for ulcerative colitis (UC).
  3. **Introduction Expansion:** The Introduction has been expanded to include references to similar studies involving probiotics in UC, providing a more comprehensive overview of the research landscape in this area.
  4. **Introduction Clarification:** We have elaborated in the Introduction on how the tested cytokines are intricately involved in UC pathogenesis, emphasizing the role of arginine in the context of UC.
  5. **Methods Clarification:** In line 49, "pasteurized" has been corrected. Additionally, we have clarified that the study involved viable *L. plantarum* bacteria. We also clarified the use of mesalazine in UC treatment.
  6. **Italicization of Species Names:** All species names have been italicized throughout the text in compliance with the specified formatting guidelines.
  7. **Figures:** We have addressed the issue of blurriness in Figures 2D and 3, ensuring that they are now clear and readable, even upon zooming. Detailed descriptions have been added to the legends of Figures 3 and 12 for improved clarity.
  8. **Taxonomic Naming:** We have changed instances of "Lactobacillus" to "Lactiplantibacillus" throughout the text to reflect the updated nomenclature.
  9. **Discussion Clarification:** We have added an explanation in the Discussion section (line 469) elucidating how arginine affects UC. The mode of action is now clarified for readers.
  10. **Optional Experiment:** While the suggestion to include separate control groups with bacteria only and goji berry only is insightful, due to the scope of this study, we were unable to conduct these additional experiments. However, we acknowledge the potential value of such an approach for future investigations.

We sincerely appreciate Reviewer X's input, which has significantly strengthened the quality and relevance of our research. Should further clarification be needed, please do not hesitate to contact us. Thank you for your invaluable contribution to our work.

Reviewer 3 Report

Comments and Suggestions for Authors

The work is very interesting. The studies of the results and their discussion are impressive. I suggest changing the introduction of this work. There is a lack of necessary data on the selection of particular strains of bacteria and goji fruits. Apart from the medical aspect, there is a lack of in-depth information on this subject, which affects the overall hypothesis of the work

Author Response

Dear Reviewer,

Thank you for your valuable feedback on our manuscript.

We appreciate your positive evaluation of the engaging nature of our work, as well as the impressive analysis of the results and discussions. We are grateful for your suggestions regarding the modification of the introduction, which have been duly noted and proved highly beneficial to us. Regarding your concern about the lack of necessary data on the selection of specific bacterial strains and goji berries, we offer the following clarifications:

  1. Lactiplantibacillus plantarum NXU0011 is a high-quality probiotic strain obtained by our research group, patented in China under application number CN116555115A. It is preserved at the China General Microbiological Culture Collection Center (CGMCC NO: 26970). This strain exhibits excellent inhibitory effects against Escherichia coli and Staphylococcus aureus, along with notable antioxidant properties. Its viability in saliva, gastric juice, and intestinal fluid is high, indicating its potential as a valuable probiotic modulator.

  1. Ningxia goji berries are precious medicinal herbs and important economic crops recorded in the Chinese Pharmacopoeia. They possess outstanding traits such as salt and alkali tolerance, adaptability to desert, drought, and cold conditions. Rich in various active substances, including polysaccharides, flavonoids, carotenoids, alkaloids, and amino acids, goji berries are widely used in the development of pharmaceuticals and functional foods. They have been applied in research and development for improving gut microbiota [1-2], functional beverages [3], and treating diseases caused by various chronic inflammations [4-9].

Your insights are immensely valuable, and we are dedicated to making the necessary revisions to enhance the quality of our manuscript.

Once again, we sincerely appreciate your valuable time and feedback.

Warmest regards,

Mingxia Nie

References

[1]   B. Tian, Z. Zhang, J. Zhao, Q. Ma, H. Liu, C. Nie, Z. Ma, W. An, J. Li, Dietary whole Goji berry ( Lycium barbarum ) intake improves colonic barrier function by altering gut microbiota composition in mice, Int J of Food Sci Tech. 56 (2021) 103–114. https://doi.org/10.1111/ijfs.14606.

[2]   P. Cremonesi, G. Curone, F. Biscarini, E. Cotozzolo, L. Menchetti, F. Riva, M.L. Marongiu, B. Castiglioni, O. Barbato, A. Munga, M. Castrica, D. Vigo, M. Sulce, A. Quattrone, S. Agradi, G. Brecchia, Dietary Supplementation with Goji Berries (Lycium barbarum) Modulates the Microbiota of Digestive Tract and Caecal Metabolites in Rabbits, Animals. 12 (2022) 121. https://doi.org/10.3390/ani12010121.

[3]   X. Dong, J. Qi, K. Xu, B. Li, H. Xu, X. Tian, H. Lei, Effect of lactic acid fermentation and in vitro digestion on the bioactive compounds in Chinese wolfberry (Lycium barbarum) pulp, Food Bioscience. 53 (2023) 102558. https://doi.org/10.1016/j.fbio.2023.102558.

[4]   H. Xie, P. Gao, Z.-M. Lu, F.-Z. Wang, L.-J. Chai, J.-S. Shi, H.-L. Zhang, Y. Geng, X.-J. Zhang, Z.-H. Xu, Changes in physicochemical characteristics and metabolites in the fermentation of goji juice by Lactiplantibacillus plantarum, Food Bioscience. 54 (2023) 102881. https://doi.org/10.1016/j.fbio.2023.102881.

[5]   Y. Hu, Effect of combined ultrasonic and enzymatic assisted treatment on the fermentation process of whole Lycium barbarum (goji berry) fruit, Food Bioscience. (2023).

[6]   K. Chien, C. Horng, Y. Huang, Y. Hsieh, C. Wang, J. Yang, C. Lu, F. Chen, Effects of Lycium barbarum (goji berry) on dry eye disease in rats, Mol Med Report. (2017). https://doi.org/10.3892/mmr.2017.7947.

[7]   Y. Liu, L. Liu, J. Luo, X. Peng, Metabolites from specific intestinal bacteria in vivo fermenting Lycium barbarum polysaccharide improve collagenous arthritis in rats, International Journal of Biological Macromolecules. 226 (2023) 1455–1467. https://doi.org/10.1016/j.ijbiomac.2022.11.257.

[8]   Y. Tian, T. Xia, X. Qiang, Y. Zhao, S. Li, Y. Wang, Y. Zheng, J. Yu, J. Wang, M. Wang, Nutrition, Bioactive Components, and Hepatoprotective Activity of Fruit Vinegar Produced from Ningxia Wolfberry, Molecules. 27 (2022) 4422. https://doi.org/10.3390/molecules27144422.

[9]   S. Lee, S. Jeong, Y. Park, H. Seo, C. You, U. Hwang, H. Park, H. Suh, Supplementation of non-fermented and fermented goji berry (Lycium barbarum) improves hepatic function and corresponding lipid metabolism via their anti-inflammatory and antioxidant properties in high fat-fed rats, Appl Biol Chem. 64 (2021) 70. https://doi.org/10.1186/s13765-021-00642-1.